# Comparison of Freshwater Mussels *Unio tumidus* and *Unio crassus* as Biomonitors of Microplastic Contamination of Tisza River (Hungary)

Wael Almeshal [1] , Anita Takács [2], László Aradi [3], Sirat Sandil [4] , Péter Dobosy [4,5] and Gyula Záray [4,5,*]

[1] Doctoral School of Environmental Science, Eötvös Loránd University, H-1053 Budapest, Hungary
[2] Department of Soil Science, Institute of Environmental Sciences, Hungarian University of Agriculture and Life Sciences (MATE), Gödöllő, Páter Károly street 1, H-2100 Budapest, Hungary
[3] Lithosphere Fluid Research Laboratory, Department of Petrology and Geochemistry, Institute of Geography and Earth Sciences, Faculty of Science, Eötvös Loránd University, H-1053 Budapest, Hungary
[4] Institute of Aquatic Ecology, Centre for Ecological Research, H-1113 Budapest, Hungary
[5] National Laboratory for Water Science and Water Security, Institute of Aquatic Ecology, Centre for Ecological Research, H-1113 Budapest, Hungary
* Correspondence: zaray.gyula@ecolres.hu

**Abstract:** *Unio crassus* and *Unio tumidus* mussels were collected at four sampling sites in the Tisza River (Hungary) to investigate their applicability as sentinel species for the biomonitoring of microplastic contamination. Since mussels, as filter feeders, are able to ingest particles only below a physically defined size, it was expected that their sentinel role in rivers is restricted to small particles, including fibers or microfibers. This assumption was confirmed by our results, as fibers were detected as the dominant particles in all the 80 mussel samples investigated. The length and diameter of the fibers changed in the size range of 20–1000 μm and 10–75 μm, respectively. The number of fibers in the individuals originating from the same sampling site was nearly two times higher in *Unio tumidus* than in *Unio crassus* and amounted to 2.7–4.9 and 5.2–8.3 items/individual. The fiber/g soft tissue ratio between these species could be characterized by a factor of three. After applying Raman spectrometry, mostly indigo-dyed polyethylene terephthalate and cellulose-based fibers, as well as a few larger (200 um) polyamide fragments, were identified. The microplastic particles stored temporarily by mussels provide only restricted qualitative information on the microplastic load of the Tisza River, and as our observations confirmed, the sampling efficiency of these 'living sampling devices' is highly species-specific.

**Keywords:** *Unionidae* mussels; biomonitoring; microplastics; microfibers; riverine environment

## 1. Introduction

Among the many adverse effects of anthropogenic activities on the aquatic environment, plastic pollution is a prominent one. While plastics are an inexpensive material with innumerable applications and provide many social benefits, they have also emerged as a persistent pollutant in the 'plastic age' due to the mismanagement of discarded plastics [1,2]. In recent years, the focus has been on microplastics (plastic particles with size < 5 mm), which are prevalent in many different environments such as seawater, riverine, sediments, soil, polar ice, and land [3–6]. Microplastics (MPs) have a variety of shapes (spherical, angular, fragment, pellet, sheet), but a large proportion of MPs in the aquatic environment are in the form of fibers arising from clothing material [1,7]. Due to their small size, a large fraction of MPs remains suspended in the water column and are easily ingested by a variety of organisms such as plankton, zooplankton, invertebrates, fishes, and mammals [3,8–11]. A number of these organisms could serve as bioindicators providing information about MP pollution in their environment; however, large-sized organisms can be opportunistically

sampled only in small numbers, unlike the smaller organisms, where the sampling of a large number of individuals is possible. Among invertebrates, mussels are important organisms that have been used to indicate pollution levels because they are sensitive to physical and chemical alterations in the aquatic environment [1,4,12–14].

Several papers in the last decade have reported the biomonitoring of MP pollution by mussels both in the marine [15–17] and freshwater [4,18–20] environments. Due to their broad geographical distribution, sessile lifestyle, easy accessibility and sampling method, tolerance to a considerable salinity range, high-stress resistance, and excessive accumulation of a wide range of pollutants, mussels are the ideal test organisms for environmental biomonitoring in the aquatic environment [12,13]. Mussels are suspension feeders (filter feeders) that are able to process large volumes (~24 L) of suspension originating from the suspended sediment daily through their filter system [21,22], but 40 L/day rates have also been reported [23]. During this filtration process, not only are phytoplankton, bacteria, and particulate organic matter ingested but also non-digestible microplastics (fragments, films, and fibers), sand, and silt particles are taken up. On the basis of the literature data, the retention time of ingested MPs in mussels varies from a few hours to 40 days, depending on their particle size and the mussel species [2]. However, the larger particles covered with mucus as pseudofeces are removed from the mantel cavity within a few hours [7]. In spite of the fact that freshwater organisms are directly affected by terrestrial run-off, wastewater, and other industrial discharges potentially containing a high level of MPs and other contaminants, ecological studies have mostly focused on marine organisms [24]. Although the use of mussels as sentinel species for large-scale monitoring programs in the marine environment has recently been recommended [25], there are more limiting factors that hamper the reliable applicability of mussels for the biomonitoring of MP contamination [26]. The most critical points are the following: (1) The capture efficiency is influenced by the size, shape, and surface properties of particles [27–30]; (2) mussels are able to sort particles based on their physical and chemical factors [30–32]; (3) differentially sized MPs are retained differently in the digestive tract of mussels [2]. The ingestion of suspended particles in the mussel–particle relationship has been widely studied during the last 40 years; however, the characterization of the mussel's habitats, namely that these animals serve as 'living sampling devices', is less investigated. Freshwater mussels, e.g., members of the family *Unionidae*, are partly embedded into the bottom sediment, and as suspension-feeding organisms, they ingest living (bacteria, algae, and protozoans) and non-living (amorphous organic matter, detritus, and inorganic mineral) particles, and simultaneously, the MPs from the suspended sediment streaming above the bottom of the riverbed. According to the published data of eight research groups, the bivalves (mussels, clams, and oysters) collected from different rivers primarily contained fibers [1,3,4,33–37]. Their length and number /individual values changed in the range of 20–1000 μm and 0–142 (Table 1). This means that the ingestion of natural or synthetic fibers is preferred to particles with other shapes, and these fibers have a longer residence time in the organisms.

**Table 1.** A summary of abundance and characteristics of microplastics in bivalves from riverine environment.

| Study Area | Bivalve Type | Abundance | Dominant Shape | Size | Dominant Color | Chemical Composition | Ref |
|---|---|---|---|---|---|---|---|
| Grand River, Canada | Mussel Lasmigona costata | 0–0.16 items/g 0–7 items/ind. | Fragments | 21–298 μm | White | PP-co-EP PP | [1] |
| Höje River, Sweden | Mussel *Anodonta anatina* | 4–142 plastic fibers/ind. | Fibers | N/A | Black | N/A | [33] |
| Saint John River, Canada | Mussel *M. margaritifera L* | 0–0.6 microfibers/g 0–10.9 microfibers/g | Microfibers | >100 μm | Blue | N/A | [31] |
| St. Lawrence River, USA | Mussel *Dreissena polymorpha D. bugensis* | N/A | Not found | N/A | N/A | N/A | [36] |

**Table 1.** *Cont.*

| Study Area | Bivalve Type | Abundance | Dominant Shape | Size | Dominant Color | Chemical Composition | Ref |
|---|---|---|---|---|---|---|---|
| Milwaukee River, USA | *Mussel Dreissena sp.* | 8.4 items/g 0.6 items/ind. | Fibers | N/A | Clear | Cotton natural Cellulose-based natural PET | [37] |
| Yangtze River, China | Asian clam *Corbicula fluminea* | 0.3–4.9 items/g 0.4–5.0 items/ind. | Fibers | 250–1000 μm | Blue | PET 33% PP 19% PE 9% | [4] |
| Thames River, UK | Asian clam *Corbicula fluminea* | 0–24 items/ind. | Fibers | N/A | N/A | PP 57% PE 9% Nylon 8% Polyallomer 8% PVP 6% Others 12% | [35] |
| Pearl River, China | Oyster Saccostrea cucullata | 1.5–7.2 items/g 1.4–7 items/ nd. | Fibers | <100 μm | Light color | PET 34% PP 19% Pe 14% PS 8% Cellophane 8% PVC 6% Polyamide 4% Expanded polystyrene 3% | [3] |

Considering the literature data mentioned above, the high amount of plastic litter transported by the Tisza River and its tributaries from the neighboring countries, the high concentration of MP particles (3177 ± 1970 items/kg) in the bottom sediment of this river, and the dominant role of fiber contamination [38], we decided to investigate the applicability of freshwater mussels as the characteristic invertebrates of the Tisza River for the biomonitoring of fiber. For this purpose, *Unio crassus* and *Unio tumidus* belonging to the family *Unionidae* were selected. They are present in the entire European mainland inhabiting running waters of different sizes and depths, particularly channels with low shear stress and fine mineral substrate [39,40]. They have a relatively high occurrence in the studied aquatic environment. The main objective of our study was to determine whether these two mussel species grown simultaneously at the same sampling sites under the same environmental conditions provide the same analytical information on MP contamination or whether the number of MPs accumulated in these mussels is actually species-dependent.

## 2. Materials and Methods

### 2.1. Sampling

Along the Tisza River, four sampling sites at the settlements of Tímár (1), Tokaj (2), Csongrád (3), and Szeged (4) were selected, which are located at river km of 552, 544, 244, and 160, respectively (Figure 1). The choice of these sampling sites provided an opportunity to study the potential effect of the tributaries (Bodrog, Sajó, Zagyva, Körös, Maros) on the MP load. These rivers flow through industrial and agricultural areas, and more small WWTPs as potential contamination sources are located at their banks. The mussels were collected during a 3-day campaign in August 2021, within a 5–10 m coastal strip and at a depth of 0.8–1.2 m, by applying a stainless steel trowel and a long-handled deep net with a brass mesh. From about 60–80 mussels at all the sampling sites, 10 *Unio crassus* and 10 *Unio tumidus* mussel species of nearly similar sizes were selected, rinsed with distilled water, placed in an aluminum foil, transferred to the laboratory on ice, and then stored at −20 °C until analysis. The other mussels were placed back into the river.

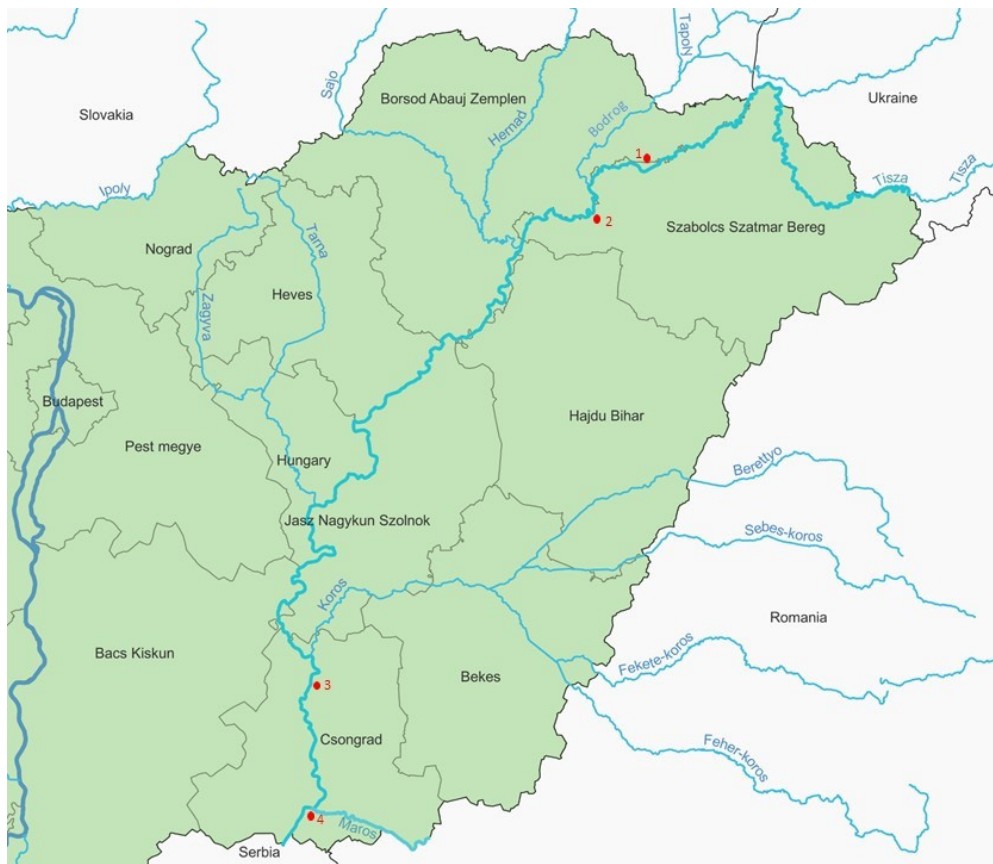

**Figure 1.** Hydrographic map of Tisza River and its tributaries arriving from Slovakia (Bodrog, Sajó) and Romania (Körös, Maros). The four sampling sites located at settlement Tímár (**1**), Tokaj (**2**), Csongrád (**3**), and Szeged (**4**) are marked with red circles.

## 2.2. Chemicals and Reagents

A Wasserlab Automatic unit (Labsystem Ltd. Budapest, Hungary) was used for the production of ultrapure water (resistivity 18.2 MΩ cm$^{-1}$). For the production of a 10% KOH solution, appropriate amounts of solid KOH granulates (VWR International, LCC) were dissolved in ultrapure water. A particle-free ZnCl$_2$ solution with a density of 1.5 g cm$^{-3}$ was prepared by dissolving solid ZnCl$_2$ (VWR International, LCC, Radnor, PA, USA) in ultrapure water, which was then filtered in a laminar box by applying a Whatman GF/C glass fiber filter with a diameter of 47 mm and pore size of 1.2 μm and a LABORPORT vacuum pump unit (KNF Lab, Freiburg, Germany).

## 2.3. Sample Preaparation

To avoid any contamination of the samples from the air, all the steps of sample preparation were carried out in a laminar box (AC2-4G8 Airstream$^{®}$ Class II, Thermo Fischer Scientific, Waltham, MA, USA). The shell lengths of the defrosted mussels were measured, and then all of the soft tissues were removed from the shells using a steel spoon and placed separately into 750 mL glass beakers, covered with a watch glass. After measuring their weights, the wet mass of the soft tissues was calculated. Based on the reported literature data for the digestion of soft tissues, 10% KOH was deemed the most suitable among the other commonly used chemicals (H$_2$O$_2$, HNO$_3$, KOH, proteinase K, and trypsin) [14,41]. The tissues (6–12 g) were separately digested in 150 mL 10% KOH at 40 °C for 24 h and incubated at room temperature for another 24 h. In order to increase the efficiency of this digestion procedure, as a first step, an ultrasonic treatment at a frequency of 37 kHz was applied for 30 min. To separate the MPs from inorganic or organic residues, a 400 mL particle-free ZnCl$_2$ solution with a density of 1.5 g cm$^{-3}$ was added to the samples,

and the mixture was covered with an aluminum foil. After the 48 h long density separation procedure, a 200 mL supernatant was filtered by applying pressure filtration with high purity synthetic air (4 bar) in order to reduce the risk of contamination from laboratory air. The particles and fibers were concentrated on a Whatman GF/C glass fiber filter with a diameter of 21 mm and pore size of 1.2 μm. Following filtration, the wet-loaded filters were placed on Petri dishes lined with crumpled aluminum foil and covered with glass lids. The filters were then dried at 60 °C in a laboratory oven, to obtain a constant weight, and stored in airtight Petri dishes at room temperature until the analysis. The procedure of blank samples was carried out using the same experimental conditions as mentioned above.

### 2.4. Analysis of Residues via Optical Microscopy and Raman Spectrometry

The loaded filters were visually inspected under a Nikon SMZ1000 stereomicroscope and a Nikon ECLIPSE LV100 POL (Nikon, Tokyo, Japan) polarization microscope with maximum magnifications of 80× and 1000×, respectively. For the chemical identification of particles and fibers, a Horiba Jobin Yvon (JY) LabRAM HR 800 Raman microspectrometer equipped with a frequency-doubled Nd-YAG green laser with a 532 nm excitation wavelength was applied, displaying 120 mW at the source and 23 mW on the sample surface. An OLYMPUS 100 × (N.A. = 0.9) objective was used to focus the laser beam on the analyzed sites. For the spot Raman analysis, a 100 μm confocal hole, with 600 grooves/mm optical grating and a cumulative 60 s exposition time, was selected. The spectral resolution of measurements varies from 2.4 to 3.0 cm$^{-1}$. The spot Raman data were processed through LabSpec 6 software 6.5.1.24 (Horiba Scientific, Paris, France).

### 2.5. Statistical Analysis

Figures were drawn with Excel, while statistical analyses were performed with R statistical software [42]. Exact Poisson tests (C-test with the poisson.test function of the 'stat' package) and Poisson regression model (with the glm function of the 'stat' package) were used to compare the mean number of fibers found in the mussel individuals of different species and habitats. The Bonferroni–Holm correction was used to avoid the problems of multiple testing [43] (with the p.adjust function of the 'stat' package).

## 3. Results

The length of the shells and the wet mass of the soft tissues of *Unio crassus* and *Unio tumidus* mussels collected at the four different sampling sites are listed in Table 2. The average length of the mussel shells (mm) was higher for *Unio tumidus* than for *Unio crassus*, but an opposite trend was observed for the soft tissue wet weight (g). The numbers of particles per individual and per gram of the soft tissues found in these mussels are demonstrated in Figures 2 and 3. Despite the higher soft tissue mass in *Unio crassus* compared with that in *Unio tumidus*, the number of fibers and fragments in individuals collected from the same sampling sites showed a different picture. The particles including the synthetic and natural fibers had a length and diameter of 20–1000 μm and 10–75 μm, respectively, and the fragments had about two times higher concentration in *Unio tumidus* than in the other mussel species ($p < 0.001$, exact Poisson test). During the investigations of loaded filters using a stereomicroscope, nine fibers of different colors were found in the following proportion: blue (54.3%), black (22.2%), gray (10.6%), brown (3.8%), red (3.4%), white (3.4%), turquoise (1.3%), green (0.6%), and pink (0.4%). Based on the Raman spectra of the reference samples, the indigo dye (Figure S1) and polyethylene terephthalate were identified as the basic materials of blue fibers in most cases (Figure S2). Among the cellulose-based fibers, two groups could be distinguished with diameters of 10–25 and 30–75 μm. The finer microfibers are characteristic of yarns made with a blend of polyester/cellulose. The thicker fibers with lengths of >200 μm are presumably the basic material of sacks used for packaging agricultural products (Figure S3). Figure S4 demonstrates a relatively large (length ~800 μm and diameter 100–120 μm) polyamide particle. It is likely that this large

fragment, as pseudofeces, was even present in the mantle cavity during the sampling. This type of MP contaminant was detected in only 2 of the 80 mussels investigated.

**Table 2.** Mean values and standard deviations of shell lengths, wet weight of soft tissues of *Unio crassus* and *Unio tumidus* mussels (n = 10), and the numbers of fibers found in these bivalves related to individuals or 1 g soft tissue. Letters (i.e., [a], [b], [c], [d]) indicate the results of the statistical analysis. Values sharing the same letters are not different significantly.

| Mussel | Site | Shell Length (mm) | Soft Tissue Wet Weight (g) | Number of Fibers/ind | Number of Fibers/g |
|--------|------|-------------------|----------------------------|----------------------|--------------------|
| *Unio crassus* | Tímár | 62 ± 5 | 10.94 ± 1.75 | 2.8 ± 0.5 [a] | 0.25 |
| | Tokaj | 68 ± 3 | 9.85 ± 2.34 | 2.7 ± 0.5 [a] | 0.27 |
| | Csongrád | 64 ± 7 | 9.53 ± 2.72 | 4.9 ± 1.2 [bc] | 0.51 |
| | Szeged | 71 ± 4 | 12.82 ± 2.12 | 3.8 ± 0.8 [ab] | 0.29 |
| *Unio tumidus* | Tímár | 67 ± 5 | 7.15 ± 1.74 | 5.2 ± 1.4 [bcd] | 0.72 |
| | Tokaj | 63 ± 6 | 6.84 ± 1.33 | 6.0 ± 1.3 [cd] | 0.87 |
| | Csongrád | 70 ± 8 | 6.94 ± 2.18 | 7.2 ± 1.9 [d] | 1.03 |
| | Szeged | 77 ± 8 | 7.95 ± 2.33 | 7.1 ± 2.4 [cd] | 0.89 |

Procedure blank 0.45 fibers/individual.

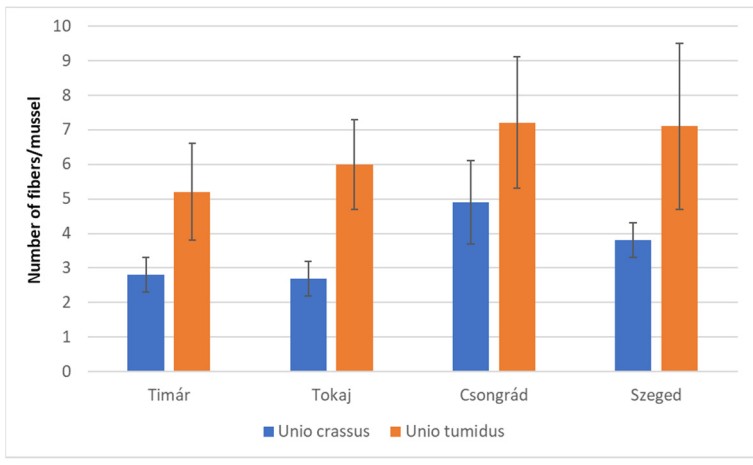

**Figure 2.** Number of fibers in mussels (n = 10) of *Unio tumidus* and *Unio crassus* collected at settlements of Tímár, Tokaj, Csongrád, and Szeged. The standard deviations are marked on the bars.

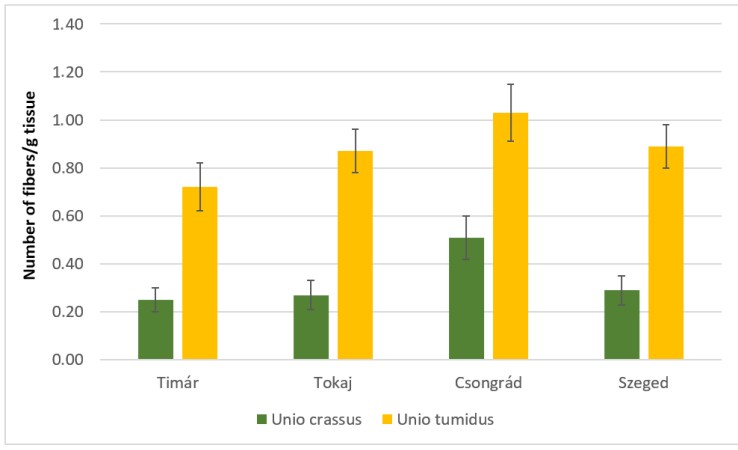

**Figure 3.** Number of fibers/g soft tissue (n = 10) of *Unio tumidus* and *Unio crassus* collected at settlements of Tímár, Tokaj, Csongrád, and Szeged. The standard deviations are marked on the bars.

## 4. Discussion

The length of the shell depends on the age of the mussel species. In the majority of *Unionoideans*, the greatest shell growth occurs in immature individuals during the first 4–6 years of life. For example, if the shell length in the case of *Unio tumidus* amounts to 60, 70, or 80 mm, the age of this mussel is about 4, 5, or 10 years [44]. This implies that the *Unio tumidus* mussels in our study with a shell length of 63–77 mm were about 4.5–5.5 years old. Comparing the wet mass of the soft tissues of these mussel species simultaneously collected at the same sampling sites, it can be established that in all the cases, the *Unio crassus* had 1.4–1.6 times higher soft tissue mass than *Unio tumidus*. This can be explained by having a higher filtration rate (3.3–4.1 L/h) than *Unio tumidus* (2.1–2.4 L/h), which results in higher nutrient transport and the faster growth of organisms [45].

The anomaly in the soft tissue mass and particle accumulation might be explained by other varying factors, such as seasonality in growth, reproduction, and feeding behavior (filtration, rejection, and egestion) [37]. Higher MP concentration in smaller-sized mussels was also reported in *Dreissena polymorpha* [46]. The authors attributed this higher MP concentration to the fact that the higher relative feeding activity, higher relative pumping rates, and larger relative gill area of smaller mussels enable them to take up more MPs per body mass. The mussel condition index (mussel dry mass/shell length) provides a gross assessment of mussel health. The mussels near potential MP sources (urban rivers and treated wastewater) are expected to have lower condition indices than those found in other sites. It was reported that the condition index could be related to food availability or habitat conditions but was not related to the MP concentration in the mussels [37]. Similarly, Wardlaw and Prosser (2020) also did not find a statistically significant relationship between the shell length and the MP abundance ($p = 0.09$) or between the soft tissue wet mass and MPs ($p > 0.5$) in the freshwater mussel *Lasmigona costata* [1]. They concluded that there was not a discernible relationship between the size and length of mussels and the number of MPs in their study, which could be due to the low number of MPs. In contrast, Berglund et al. (2019) reported a positive correlation between the size of the mussel and the number of MP fibers accumulated by them. They explained that larger mussels filter larger volumes of water and, thus, accumulate more fibers [33].

The measurement of MPs in mussels represents the internal exposure level of MPs to these organisms and can help in evaluating the ecological risks due to MP uptake [4]. Comparing the literature data related to the riverine environment (Table 1) with our results, it can be established that the abundance of fibers and fragments in the mussels, clams, and oysters collected from different rivers changed in the range of 0–142 items/individual, while in our case, the calculated mean values using 10 samples from each sampling site amounted to 2.7–4.9 and 5.2–8.3 items/individual for *Unio crassus* and *Unio tumidus*, respectively. These values were obtained after subtracting the procedure of blank samples (0.45 items/individual). This implies that the species *Unio tumidus* is a more efficient 'sampling device' to study the fiber contamination in freshwater rivers with moderate velocities, compared with *Unio crassus*. A few other studies have reported the MP abundance in mussels in relation to the mass of the mussels: 0–0.16 particles/g wet weight in *Lasmigona costata* [1], 0.3–4.9 MP/g wet weight in Asian clams (*Corbicula fluminea*) [4], 1.5–7.2 items/g wet weight in the oyster *Saccostrea cucullate* [3], and 0.35–0.38 fibers/g wet weight in *Mytilus sp.* [12]. Our data changed in the range of 0.25–0.51 and 0.72–1.03 items/g soft tissue for *Unio crassus* and *Unio tumidus*, respectively. However, in the literature, there are no experimental data on the comparison of the stored amount of MPs in the different mussel species grown under the same environmental conditions. Schessl et al. (2019) conducted an experiment to study the microbead content in *Dreissena polymorpha* and *Dreissena bugensis* grown in the littoral zone of the Upper St. Lawrence River in the USA; however, they did not detect any MP particles in the mussels. Their results were explained by the relatively small size of the collected mussels (an average length of 17.33 mm) and the limited ingestion capability of these organisms [34].

When evaluating the fiber content of the mussels in the direction of flow from Tímár to Szeged, only a slow increment in fiber concentration could be observed, except for the sampling site at Csongrád (Table 2), where the stream velocity was extremely low (less than 0.1 m/s) in a small bay, and the bottom of the bed was muddy. If we compare the data obtained from the sampling sites Tímár and Szeged, where the hydrological conditions were similar, it can be established that, due to the transport of contaminants between the tributaries (Bodrog, Sajó, Zagyva, Körös, Maros), along the 392 km section of the Tisza River, the number of fibers/g soft tissue values increased by 35% in both mussels species. Our observations are in agreement with the experimental data of Kiss et al. (2020), who observed about 20% higher MP amount in the sediments of these tributaries compared with the amount found in the main river [38].

The frequent presence and predominance of fibers in mussels [3,4,33] are not surprising if we consider that fibers make up to 64–100% of MP contaminants in the water phase of several rivers, e.g., Ciwalengke [47], Yellow [48], and Antuã [49]. The high occurrence of fibers in mussels could be due to mussel ecology and the high amount of fibers discharged into the rivers from wastewater treatment plants. Microfibers primarily arise from washing clothes. De Falco et al. (2019) demonstrated that the microfibers released during washing range from 124 to 308 mg kg$^{-1}$ of the washed fabric, and depending on the type of washed garment, it would correspond to a number of microfibers ranging between 640,000 and 1,500,000. The most abundant fraction of the microfibers shed was retained by filters with a pore size of 60 μm and presented an average length of 360–660 μm and an average diameter of 12–16 μm [50]. The fibers, particularly the smaller ones, are not removed by the currently available wastewater treatment technologies and, thus, accumulate in the aquatic environment [3,4,33].

Similar to our observations, the blue-colored fibers were predominant in the bivalves collected from St. John and Yangtze rivers [4,36]. However, in other studies, the white-colored, light-colored, and transparent fibers were also dominant [3,37,51]. Since mussels are not able to selectively uptake fibers based on color, the observed proportion of colors likely represents a real picture of the proportion of dyed fibers in the ingestible size range. However, it is worth noting that the dyes affect the surface properties of the fibers and, thus, the biofilm formation on their surface. This can increase the specific gravity, thus causing a change in the depth distribution of the fibers in the water body of rivers. A debatable idea was put forward by Berglund et al. (2019) that mussels could have a color preference for food, and the color distribution of MPs in mussels could be a result of both the color preference of the mussels and the dominant color in the water [33]. The color of the MPs ingested by mussels was also reported to be dependent on the season. In autumn and summer, the bivalves ingested more transparent MPs, while in winter, they ingested more blue MPs, and in spring, they contained an equal amount of both MPs [13].

The rejection of larger particles was demonstrated by Ward et al. (2019) in the mussel *Mytilus edulis*, where the mussel rejected a lower number of 19–113 μm sized MP spheres and a significantly higher number of 1000 μm sized MP spheres [7]. The proportion of the MP spheres rejected in pseudofeces increased with an increase in the sphere size, while for the fibers, the rejection was variable and displayed no trend with regard to size. The ability of mussels to size-select is due to the presence of two digestive paths (intestinal and glandular path) and the microstructures in their digestive tract. The immediate bulk egestion of large MPs could occur through the intestinal path, while the longer retention of smaller MPs (1–10 μm) could occur through the glandular path [2].

Based on our observations, it can be established that, under the environmental conditions of the Tisza River, *Unio tumidus* is a more efficient sample than *Unio crassus* to characterize the fiber contamination of rivers and follow their concentration changes during long-term monitoring.

## 5. Conclusions

The dominant particles in both mussel species collected in the Tisza River were fibers, like in other bivalves taken from different rivers, e.g., Thames in the UK, Yangtze and Pearl in China, St. John in Canada, Milwaukee in the USA, and Höje in Sweden. It should be noted, however, that, as a living sampling device, *Unio tumidus* has a higher accumulation capacity than *Unio crassus*; therefore, this species can be recommended as a biomonitor in this catchment area to study the changes in fiber contamination. If we take into account the expected development of world fiber production until 2030 [52,53], polyester production will significantly increase, while in the case of natural fibers (cotton, wool, cellulosic), only slight growth or stagnation is expected. Since the wastewater treatment plants located at the bank of rivers are the main sources of fiber contaminants in the riverine environment, we need to calculate the increasing emission levels of polyester fibers in rivers and simultaneously the higher concentration of these contaminants in suspension-feeding organisms. As far as the issue of the biomonitoring of the different MP contaminants is concerned, the sampling with living organisms (biomonitoring) suffers from several limiting factors, as published by Ward's and Hollein's groups [26,37]. To obtain reliable quantitative data on the total MP load of rivers, instead of biomonitoring, an automated sampling system should be applied that collects all the streaming-suspended particles in a large size range at different depths, and thus, a complete and more reliable picture can be formed on the MP load of rivers following the appropriate analytical investigations of particles.

**Supplementary Materials:** The following supporting information can be downloaded at: https://www.mdpi.com/article/10.3390/environments9100122/s1, Figure S1: Stereomicroscopic picture and Raman spectrum of blue fiber found in *Unio tumidus* mussel collected at sampling site Szeged and Raman spectrum of indigo dye reference material; Figure S2: Stereomicroscopic picture and Raman spectrum of a fiber found in *Unio tumidus* mussel collected at sampling site Szeged and Raman spectrum of PET reference material. Figure S3: Stereomicroscopic picture and Raman spectrum of a thick fiber found in *Unio tumidus* mussel collected at sampling site Szeged and Raman spectrum of cellulose reference material. Figure S4: Stereomicroscopic picture and Raman spectrum of a fragment found in *Unio tumidus* mussel collected at sampling site Szeged and Raman spectrum of polyamide reference material.

**Author Contributions:** Conceptualization, G.Z., methodology, L.A., A.T. and P.D.; formal analysis, P.D.; investigation, W.A., S.S., P.D., A.T. and L.A.; data curation, W.A.; writing—original draft preparation, W.A.; writing—review and editing, G.Z. and P.D.; visualization, P.D. and L.A.; supervision, G.Z. All authors have read and agreed to the published version of the manuscript.

**Funding:** The research presented in the article was carried out within the framework of the Széchenyi Plan Plus program with the support of the RRF 2.3.1 21 2022 00008 project.

**Data Availability Statement:** Not applicable.

**Acknowledgments:** The authors thank József Szekeres for selecting the sampling sites and collecting the mussel samples. Wael Almseshal and Sirat Sandil thank the support of the Stipendium Hungarcium Foundation.

**Conflicts of Interest:** The authors declare no conflict of interest.

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
