# Peer review of "Comparison of Freshwater Mussels Unio tumidus and Unio crassus as Biomonitors of Microplastic Contamination of Tisza River (Hungary)"

_environments, doi:10.3390/environments9100122_

Round 1

Reviewer 1 Report

This paper investigates the microplastic pollution in Tisza River applying as biomonitors the mussels Unio tumidus and Unio crassus. This paper is interesting scientific. However, some issues in the manuscript need improvement. I suggest the publication of the manuscript after minor revisions.

Q1

What are the special characteristics of the sampling points? what are the reasons chosen from the total length of the river (392km)? why is an opportunity to study the potential effect of tributaries (Bodrog, Sajó, Zagyva, KÅ‘rös, Maros) on the MP load (Lines 105-106)? The reason? At line 306, generally described that “Since the wastewater treatment plants located at the bank of rivers…”

Q2

The mussels were collected during a 3-day campaign in August 2021 (Line 106). During the investigations of loaded filters by stereomicroscope, nine fibers of different colors were found in the following proportion: blue (54.3%), black (22.2%), gray (10.6%), brown (3.8%), red (3.4%), white (3.4%), turquoise (1.3%), green (0.6%), and pink (0.4%) (Lines 171-174). According to reference on Lines 280-281: “The color of MPs ingested by mussels was also reported to be dependent on the season. In autumn and summer, the bivalves ingested more transparent MPs…”

Why the majority of MPs is blue and black instead of transparent, which the reference describes?

Q3

Finally, using the mussels as indicators, can we get a specific result for the river MPs pollution load assessment?

Author Response

Thanks for your critical evaluation of our manuscript helping us to improve its quality.

Question: What are the special characteristics of the sampling points? what are the reasons chosen from the total length of the river (392km)? why is an opportunity to study the potential effect of tributaries (Bodrog, Sajó, Zagyva, KÅ‘rös, Maros) on the MP load (Lines 105-106)? The reason? At line 306, generally described that “Since the wastewater treatment plants located at the bank of rivers…”

Answer: The four sampling sites were selected to study the effect of tributaries, as mentioned in the 2.1 subchapter.

Sampling site

1. Characteristic place to determine the MP contamination of Tisza River arrived from the neighboring Ukraine and Romania

2. It was selected to study the effect of Bodrog River arriving from Slovakia and having the highest water yield among the tributaries of Tisza.

3. Appropriate sampling sites to characterize the potential effect of tributaries KÅ‘rös and Maros, respectively, arriving from Romania.

Question: The mussels were collected during a 3-day campaign in August 2021 (Line 106). During the investigations of loaded filters by stereomicroscope, nine fibers of different colors were found in the following proportion: blue (54.3%), black (22.2%), gray (10.6%), brown (3.8%), red (3.4%), white (3.4%), turquoise (1.3%), green (0.6%), and pink (0.4%) (Lines 171-174). According to reference on Lines 280-281: “The color of MPs ingested by mussels was also reported to be dependent on the season. In autumn and summer, the bivalves ingested more transparent MPs…”

Why the majority of MPs is blue and black instead of transparent, which the reference describes?

Answer: Ding et al. (2021) studied the MP-uptake by scallop (Chlamis farreri), mussel (Mytilus galloprivincialis), oyster (Cassostrea gigas) and clam (Ruditapes philippinarum). They established that the occurance of transparent and colored MP particles in these bivalves, is species specific.  For example in case of Cr.gigas the blue, while for Ch.farreri, M. galloprovincialis and R. philippinarum the transparent particles were predominant. In addition Ding et al. observed a seasonal effect in the ratio of transparent and colored MP-particles. Until now there is not clear explanation to clarify this phenomenon. Presumably the seasonal effect is also species specific and cannot be generalized.

Question: Finally, using the mussels as indicators, can we get a specific result for the river MPs pollution load assessment?

Answer: Using the Unio Tumidus and Unio Crassus as bioindicators we receive information first of all on the fiber contamination because the ingestible particle size is limited by the geometrical parameters of their filter system. Particles with diameter higher than about 8 micrometers will not be ingested by the mussels and will be removed from the bivalves as pseudofaces.

Reviewer 2 Report

The manuscript “Comparison of freshwater mussels Unio tumidus and Unio crassus as biomonitors of microplastic contamination of Tisza River (Hungary)” describes an interesting investigation on the efficiency of two different species of freshwater mussels to utilize as sentinel species for biomonitoring of microplastic contamination. Many aspects are well described, especially in the Discussion session where the meaning of the all available results have been analysed comparing them with other studies. Neverthless, the lack of statistical analysis could lead to a not completaly correct interpretation of the results. 

The supplementary material is clear and useful.

The references are appropriate and updated, but not exhaustive in some cases (see the comments in the attached file).

Some clarifications are required.

Author Response

First of all, thanks for your questions and suggestions to improve the quality of our manuscript.

INTRODUCTION

Question: Lines 74-75 - “The mussels are partly embedded into the bottom sediment…”. This statement is valid for mussels living in rivers. In the sea the savage mussels (Mytulus galloprovincialis, Mytilus edulis) are mostly settled on hard natural or artificial substrates. “Mussel” is a generic name including a broad range of species both of fresh- and marine waters. Please specify.

Answer: This is an important correction what you suggested. The sentence has been modified as follows: “The freshwater mussels e.g. the member of family Unionidae are partly embedded…..

MATERIALS AND METHODS

Question: Lines 103-106 - In the description of the sampling sites should be add some information on the main human activities carried out along the Tisza River and its tributaries to understand possible sources of pollution. Are there areas along the river with higher impact than others?

 Answer: In order to receive information on the effect of larger tributaries of Tisza River the sampling sites were selected below their estuaries. These rivers flow through industrial and agricultural areas and more small WWTPs as potential contamination sources are located at their banks. The new sentence was inserted into the text.

Question: Lines 108-109: Please, specify the material from which the trowel and the dip net are made.

Answer: A stainless steel trowel and a long-handled deep net with brass mesh were applied. These information were added to the original text.

RESULTS

Question: Lines 164-165 - On the base of the values reported in the Table 2, taking into account the standard deviation (is it standard deviation?, specify in the legend), it seems that there are not differences between the two species in terms of shell length and soft tissue in the most of cases. The same regarding the number of fibers (fig.2) found in Unio crassus. I suggest to apply statistical analysis to better interpret measures and results.

Answer: On basis of your suggestion the following new headline was written for Table 2. “Mean values and standard deviations of shell lengths, wet weight of soft tissues of Unio crassus and Unio tumidus mussels (n=10) and the found numbers of fibers in these bivalves related to individuals or 1 g soft tissue. Statistical analyses were added.

Question: Lines 188-192 - Specify in the legends what the bars are (standard deviation? standard errors..?)

Answer: The figure captions of Fig 2 and Fig 3 have been supplemented with this short information: The standard deviations are marked on the bars.

DISCUSSION

Question: Line 237 - Write Saccostrea cucullate in italics.

Answer: Saccostrea cucullate is written in italics.

SUPPLEMENTARY MATERIAL

Question: Please, improve the quality of the graphs and pictures.

Answer: Quality of the graphs and pictures was improved.

Round 2

Reviewer 2 Report

The Authors followed the Referee's suggestions and the manuscript has been improved. The paper can be published in the present form.